# Two-dimensional gradients in magnetic properties created with direct-write laser annealing

Lauren J. Riddiford [1,2,6] ✉, Jeffrey A. Brock[1,2,5,6] ✉, Katarzyna Murawska[1,2], Jacob Wisser[3], Xiaochun Huang[2,4], Nick A. Shepelin [2], Hans T. Nembach[3], Aleš Hrabec [1,2] ✉ & Laura J. Heyderman [1,2]

Across the fields of magnetism, microelectronics, optics, and others, engineered local variations in material properties can yield groundbreaking functionalities that play a crucial role in enabling future technologies. One-dimensional lateral gradients in material properties give rise to a plethora of new effects in thin-film magnetic systems. However, extending such gradient-induced behaviors to two dimensions has been challenging to realize experimentally. Here, we demonstrate the creation of two-dimensional complex patterns with continuous variations in magnetic anisotropy, interlayer exchange coupling, and ferrimagnetic compensation at the mesoscopic scale in numerous application-relevant magnetic materials. We exploit our engineered gradients in material properties to demonstrate novel magnetic functionalities, including the creation of a spin wave band pass filter and an architecture for passively resetting the position of a magnetic domain wall. Our results highlight the exciting new physics and device applications enabled by two-dimensional gradients in thin film properties.

Over the past few years, there has been an intense research endeavor to create and understand the unique effects and functionalities that emerge when material properties are varied in a gradient manner over micro- and nanoscale lateral dimensions[1-10]. Within the field of magnetism, the technological necessity of engineering complex magnetic energy landscapes has been demonstrated by recent reports on a number of novel, one-dimensional (1D) gradient-induced effects[11-16]. For example, it has been shown that the symmetry breaking originating from lateral gradients in the magnetic compensation temperature of ferrimagnets[11] or from varying the magnitude of the interlayer coupling strength of synthetic antiferromagnets[12] enables magnetic field-free switching of magnetic elements using spin-orbit torques. As all-electrical control of magnetization is necessary to realize

energy-efficient, high-density spintronic computing architectures such as magnetic random-access memory (MRAM), there is a crucial need to develop scalable techniques to enact these functionalities[17-19]. In addition, moving beyond 1D to complex multidimensional designs can yield novel devices and new physics.

Gradients in magnetic properties are also particularly relevant to unconventional, beyond-von Neumann computing schemes. For instance, localized gradient changes in the anisotropy strength of ferromagnetic materials can impact both the propagation direction and dispersion of magnons[20,21], as well as the dynamics of magnetic domain walls (DWs) as they move through the gradient[22]. Indeed, it has long been predicted that sufficiently strong gradients in perpendicular magnetic anisotropy strength should enable spontaneous, self-driven

[1]Laboratory for Mesoscopic Systems, Department of Materials, ETH Zurich, Zurich, Switzerland. [2]PSI Center for Neutron and Muon Sciences, Villigen PSI, Switzerland. [3]Applied Physics Division, National Institute of Standards and Technology, Boulder, CO, USA. [4]Laboratory of Inorganic Chemistry, Department of Chemistry and Applied Biosciences, ETH Zurich, Zurich, Switzerland. [5]Present address: Department of Physics and Energy Science, University of Colorado Colorado Springs, Colorado Springs, CO, USA. [6]These authors contributed equally: Lauren J. Riddiford, Jeffrey A. Brock. ✉e-mail: lauren.riddiford@psi.ch; jbrock8@uccs.edu; ales.hrabec@psi.ch

DW motion[23], which is relevant for DW-based in-memory and neuromorphic computing applications. However, an experimental realization of the requisite anisotropy energy landscape has remained elusive.

Typically, magnetic properties have been modified at small length scales and in intricate patterns using focused beams of ions (typically He$^+$ or Ga$^+$) and electrons, as well as thermal scanning probes[14,24–28]. However, these approaches generally require specialized equipment and ultra-high vacuum environments. Furthermore, the scanning of focused ion beams over large areas is typically slow and suffers from dose instability[29,30]. Because of these challenges, performing systematic studies becomes prohibitively time-consuming[31], and several concerns regarding the industrial scalability of these processes have yet to be addressed[32,33]. As such, developing new techniques to locally impart gradients in magnetic properties that allow for rapid iteration and industrial scalability would dramatically expand the avenues for research.

Laser annealing, in which thermal energy is imparted to a material using a focused laser beam, has been used in magnetism to modify the saturation magnetization, exchange bias, and magnetic easy axis orientation of thin films[34–36]. However, the instruments used to perform laser annealing have typically been lab-built, bespoke setups offering laser spot sizes on the order of 10 μm–20 μm without any complex, high-resolution patterning capabilities[28]. Thus, laser annealing has not shown much promise up to now as an applications-relevant technique for micro- or nanoscale devices. More recently, the suitability of direct-write laser annealing (DWLA) for achieving local, binary modifications in material properties has been demonstrated[37,38]. Because DWLA can be performed using existing laser-based photolithography equipment and does not require patterned resist layers or ultra-high vacuum environments[39], the process is significantly streamlined compared to other approaches, allowing for rapid optimization of exposure parameters that yield desirable material changes. However, complex, two-dimensional (2D) gradients in a variety of magnetic properties and materials have yet to be demonstrated using DWLA.

Here, we demonstrate the creation of complex 2D gradients in the material properties of thin films, including crystallization, oxidation, and interdiffusion, by DWLA. In turn, these local changes are used to create elaborate magnetic energy landscapes in several important magnetic thin-film systems. In particular, we locally engineer continuous variations in the magnetic anisotropy in ferromagnetic and synthetic antiferromagnetic materials, the compensation temperature of ferrimagnets, and the interlayer exchange coupling strength of synthetic antiferromagnets (SAFs), exemplifying the broad utility of DWLA. Furthermore, we demonstrate the manipulation of spin waves and magnetic domain walls with magnetic property gradients, highlighting the relevance of 2D gradients in material properties for magnonics and spintronics research and applications. These results bring new functionality and relevance to laser annealing as a technique to engineer desirable material performance.

## Results and discussion
In Fig. 1, we summarize the various physical properties that can be modified using DWLA and how they impact magnetic properties. The process of DWLA is depicted within the innermost oval of Fig. 1. The laser is raster scanned in a serpentine path (indicated by the white dashed line) over the sample, with the fluence automatically adjusted following whatever design is input, which here gives a linear gradient in fluence. At each point, the laser deposits thermal energy locally at the sample surface, which is partially absorbed by the film and dissipated through the substrate. The resulting temperature increase occurs ~100 nm into the film (see Supplementary Fig. S1), but we choose thin film

heterostructures with a total thickness <20 nm to maximize the uniformity of temperature increase through the thickness of our films. We use Si substrates with a 300 nm-thick SiOx coating because of their low thermal conductivity, which maximizes the temperature increase in the film for a given laser fluence. The larger oval encompasses the material changes we observe in different thin film systems due to DWLA, including crystallization, oxidation, and interdiffusion. Example magnetic configurations resulting from the microstructural changes are shown: gradients in the (i) magnetic anisotropy due to crystallization or interdiffusion, (ii) interlayer exchange coupling due to interdiffusion, and (iii) magnetic compensation due to oxidation.

### Controlling structural transformations with DWLA
To grasp the range of applications possible with DWLA, it is important to understand the microstructural changes enacted in different films during the annealing process. Given that the 405 nm-wavelength laser light we employ is well below the band gap of silicon, and all materials implemented here are metallic, the laser acts as a local heat source. Here, we outline the three changes in physical properties observed by laser annealing films: crystallization, oxidation, and interdiffusion (shown in the larger oval of Fig. 1). To demonstrate that DWLA is a broadly applicable technique − both in the range of physical transformations it induces and wide variety of possible materials − we examine its effect on the following material systems: (1) ferromagnetic CoFeB/MgO thin films, (2) ferrimagnetic CoGd films, and synthetic antiferromagnets composed of (3) Co/Cr, (4) Co/Ta or (5) CoFeB/Pt/Ru (details of sample compositions are given in "Synthesis"). These were chosen because researchers performing uniform annealing experiments observed one dominant physical transformation in these systems in the temperature range of 200 °C to 500 °C[40–44].

The evolution of the microstructure of CoFeB/MgO heterostructures on heating has been extensively investigated due to their use in magnetic tunnel junctions, which are ubiquitous in MRAM devices. Here, it is known that thermal annealing induces crystallization of CoFeB/MgO layers[45]. In Fig. 2a, cross-sectional transmission electron microscopy (TEM) images of two lamellae cut from a single sample are compared, where one lamella is cut from an as-grown region (upper image), and the other is cut from a region that has been laser annealed at a fluence of 2.5 J cm$^{-2}$ (lower image). The CoFeB/MgO layers in the as-grown sample are hard to distinguish from one another, and no diffraction peaks in the accompanying electron diffraction patterns are noticeable. For the lamella from the laser annealed region, there is a significant contrast difference between the two layers, suggesting a difference in density. In addition, diffraction peaks become visible for these layers, similar to the appearance of diffraction peaks observed in ultrathin CoFeB/MgO annealed in a furnace[40]. Complementary x-ray reflectivity (Supplementary Fig. S2) and magnetometry measurements (Supplementary Fig. S3) suggest minimal interdiffusion or oxidation. From this, we can conclude that the CoFeB/MgO has crystallized as a result of DWLA (TEM micrographs of larger regions of each lamella are shown in Supplementary Fig. S4). While laser annealing of CoFeB/MgO has been previously demonstrated[46], we have the new ability to tune the degree of crystallization locally.

Next, we consider the ability of DWLA to modify the chemical composition of thin films. The oxidation of ferrimagnetic CoGd films in response to DWLA was evaluated using dynamic secondary ion mass spectroscopy (SIMS). For this, an Ar$^+$ ion beam is used to gradually remove the film, layer by layer, by sputtering. The secondary ions produced during sputtering are electrostatically guided to a quadrupole detector, which filters ions based on their mass and charge. In our case, the instrument was tuned to select to O$^-$ ions. Comparing the SIMS profiles of an as-grown CoGd film with a CoGd film after exposure with a laser fluence of 2 J cm$^{-2}$ (see Fig. 2b), it becomes evident that laser annealing significantly increases the oxygen content within the

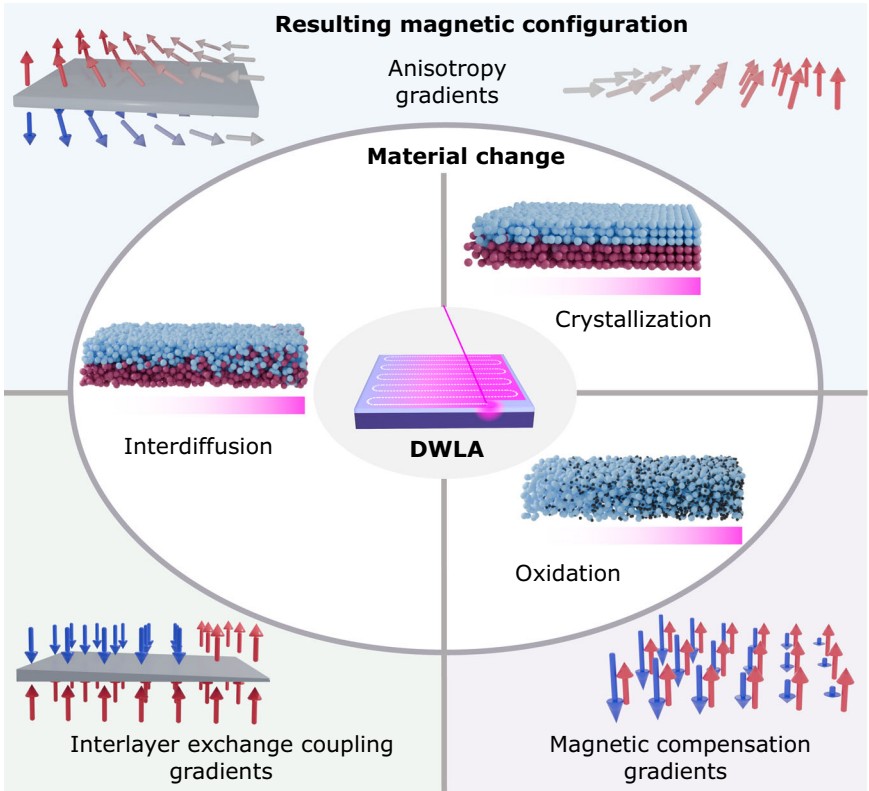

**Fig. 1 | Heat-activated physical and chemical changes provided by DWLA and their impact on the magnetic properties.** With direct-write laser annealing (DWLA), thermal energy is locally deposited in the film and dissipated by the substrate, illustrated in the schematic within the innermost oval. The laser is rastered in a serpentine pattern, and the laser fluence is adjusted as it rasters according to the design input. Here, a linear gradient in fluence, increasing from left to right, is shown. The larger oval encompasses the three material changes enacted by the laser: interdiffusion, crystallization, and oxidation. The blue and red spheres represent atoms of two different thin film layers, and the small black spheres represent oxygen. For each schematic, the material change is shown for a linearly increasing laser fluence from left to right, indicated by the increasing intensity of pink in the bar beneath the schematics. The outermost box reflects the possible configuration of the magnetic moments in the thin film systems resulting from the microstructural transformation. On the left side, there are two magnetic layers with a nonmagnetic spacer (shown in gray). At the upper right, there is a single magnetic layer. At the lower right, there is a single magnetic layer with two magnetic sublattices (given by the two differently colored arrows). Red (blue) arrows indicate the local magnetization is pointing up (down), and gray arrows lie in the film plane.

CoGd layer. The chemical changes we observe in response to DWLA in our CoGd ferrimagnets are in agreement with previous work in which the oxygen content of ferrimagnetic films was modified using an oxygen plasma, ion irradiation, or furnace annealing[41,47,48]. This change in the oxygen content in response to DWLA occurs without significantly modifying the crystal structure of the CoGd system (see Supplementary Fig. S5).

The final physical transformation we observe in thin films after laser annealing is interdiffusion, which we detect in synthetic antiferromagnets through x-ray diffraction following the method given in ref. 43. A symmetric $2\theta - \omega$ scan of a CoFeB/Pt-based synthetic antiferromagnet with a Ru interlayer in the $2\theta$ range where film peaks appear (blue curve in Fig. 2c) reveals one primary diffraction peak for as-grown films corresponding to Pt (111), as well as two small peaks corresponding to two different Ta phases. Half of the film is laser annealed at a fluence of 3.6 J cm$^{-2}$, and it can be seen that this half of the film has a different primary peak at a $2\theta$ value higher than that of the as-grown region (red curve in Fig. 2c). This new peak location corresponds to the (10$\bar{1}$) diffraction peak of several CoPt and FePt alloys. None of the peak locations match alloys with Ru (see Supplementary Fig. S6 for the XRD scan over a broad $2\theta$ range). We exclude oxidation in these heterostructures through SQUID-VSM measurements of the magnetization before and after annealing (see Supplementary Fig. S7). In addition, using a combination of x-ray diffraction measurements, SIMS measurements, and comparisons to films deliberately alloyed during growth, we find that the

Co/Cr/Co- and Co/Ta/Co-based heterostructures also experience interdiffusion in response to laser annealing without a change in their crystalline structure (see Supplementary Fig. S8). Altogether, these results demonstrate that our DWLA approach can give interdiffusion in thin metallic layers that is also seen with furnace annealing or ion irradiation[42,49].

In the following sections, we implement DWLA in magnetic thin film systems to create gradients in magnetic properties for magnonics and spintronics applications. However, these physical changes are not limited to magnetic films and can be implemented into any thin film material that undergoes a thermally-induced structural and/or chemical change. For example, phononic and photonic metamaterials have been made by nanostructuring materials[50], but creating gradients in the refractive index of a material could widen possibilities. Indeed, very recently, DWLA was used to create photonic circuits by modifying the crystal structure of a thin film in a binary manner[38]. Our gradient technique therefore dramatically expands the possibilities for photonic devices. In addition, while surface roughness gradients created using multiple chemical and microfluidic patterning steps have been used to control the properties of cultured cells[51], DWLA offers a path to obtain similar effects through a substantially more straightforward fabrication process.

**Tuning magnetic properties with DWLA**

Having demonstrated that DWLA enables a variety of physical and chemical changes in thin film systems, we now link these physical

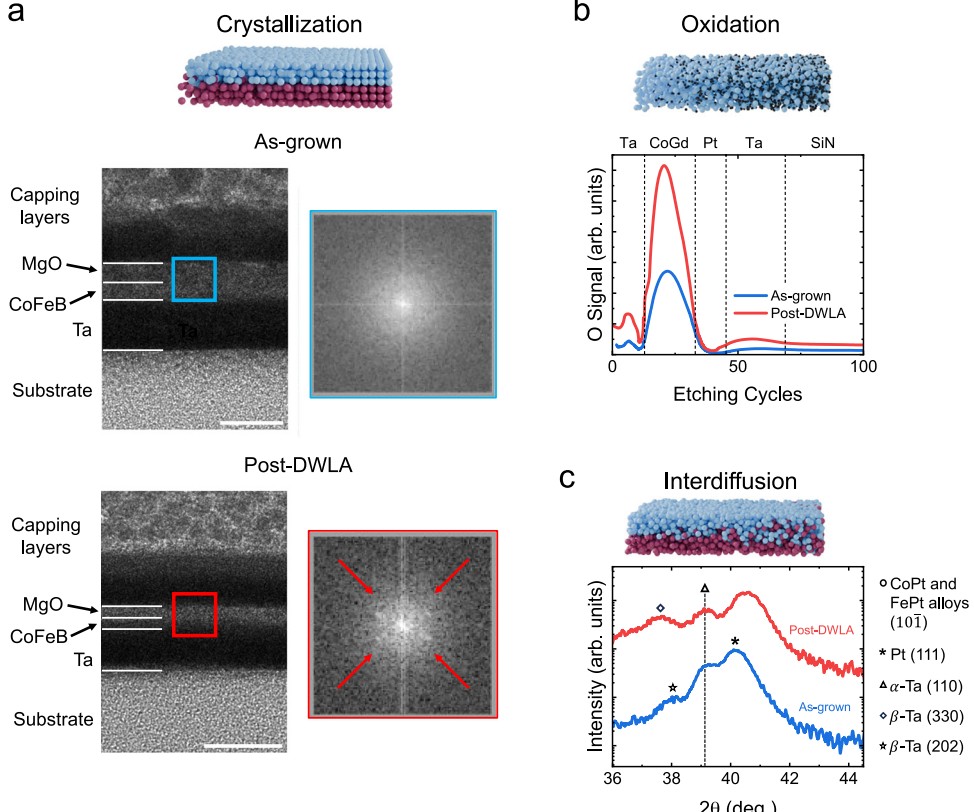

**Fig. 2 | Chemical and physical changes with DWLA. a** Crystallization: cross-sectional transmission electron microscopy images (left) of CoFeB/MgO before (upper) and after (lower) exposure with DWLA. Film layers are labeled; below the Ta is the SiOx substrate, and above the MgO is a Ta cap and a Pt/carbon protective layer. The colored boxes in each TEM image show the region of the film where the electron diffraction patterns (right) were recorded. While prior to annealing, the CoFeB and MgO have similar densities with no diffraction peaks visible, after annealing, there is a visible difference in density, and four diffraction peaks appear (indicated by the red arrows), suggesting that crystallization has occurred. Scale bar = 10 nm. **b** Oxidation: oxygen sensitive SIMS profiles for CoGd films in the as-grown state (blue) and after uniform laser exposure with a fluence of $2\,J\,cm^{-2}$ (red). The etching cycles sequentially reveal the amount of oxygen in the Ta, CoGd, Pt, and Ta layers, and the SiN substrate, bounded by vertical dashed lines. **c** Interdiffusion: $2\theta - \omega$ x-ray diffraction scans (with $\omega$ fixed to the same value as $\theta$) of a CoFeB/Pt-based synthetic antiferromagnet with a Ru spacer before (blue) and after (red) laser exposure. See "Synthesis" for specific sample composition and structure.

changes to changes in the magnetic properties. To characterize the ways in which DWLA affects the magnetic properties of different materials systems, we measure areas annealed with different laser fluences across a single film (see "Magneto-optic Kerr effect measurements" and "Magnetic characterization" for more details). This allows us to record small changes in magnetic properties due to increased laser fluence without having to account for variations in growth conditions, and therefore the magnetic properties, that often exist between samples grown at different times.

In the case of CoGd transition metal-rare earth ferrimagnets, we find that the oxidation facilitated by DWLA, shown in Fig. 2b, provides a means to tune the dominant magnetic sublattice at a given temperature (i.e., whether the Co magnetization aligns antiparallel or parallel to the applied magnetic field). For laser fluences $<1.3\,J\,cm^{-2}$, the coercivity at room temperature increases with increasing fluence, while for fluences $>1.3\,J\,cm^{-2}$, the coercivity decays with increasing fluence (Fig. 3a). From polar MOKE (pMOKE) hysteresis loops (see Supplementary Fig. S9a), we observe that perpendicular magnetic anisotropy is maintained over the range of laser fluences used as can be seen from the consistent squareness of the hysteresis loops. Furthermore, we find that the sign of the Kerr rotation measured in our hysteresis loops changes from positive to negative as the laser fluence is increased across the point where the coercivity diverges (shown for laser fluences of $0.8\,J\,cm^{-2}$ and $1.6\,J\,cm^{-2}$ in Fig. 3a and for all fluences in Supplementary Fig. S9a). Given the configuration of our MOKE microscopy platform (see "Magneto-optic

Kerr effect measurements" for details), this indicates that laser annealing changes the magnetically dominant sublattice. The divergence in the coercivity and the change in the sign of the Kerr rotation shown in Fig. 3a are characteristic of ferrimagnets in the vicinity of a change in the dominant magnetic sublattice. This can be the result of a change in thickness, temperature, proportion of TM to RE atoms, ion irradiation dosage, or, as in our case, oxidation[44,47,52–54]. The change in the degree of ferrimagnetic compensation revealed by our MOKE characterization is further confirmed by magnetometry measurements of continuous films uniformly exposed using various laser fluences (see Supplementary Figs. S9b and S9c). Here we observe a net magnetization of zero in the vicinity of the fluence where the coercivity diverges in Fig. 3a ($-1.3\,J\,cm^{-2}$), indicating that the magnetization of the Co and Gd sublattices is of equal magnitude and opposite orientation, as expected at ferrimagnetic compensation.

We now turn to CoFeB/MgO-based heterostructures, where crystallization of the CoFeB/MgO layers due to laser annealing, shown in Fig. 2a, can yield strong perpendicular magnetic anisotropy (PMA) resulting from hybridization of the Fe-O orbitals. To demonstrate this change in anisotropy, CoFeB/MgO films are grown with in-plane magnetic anisotropy. As areas of the film are laser annealed with increasing fluence, the effective magnetic anisotropy field $\mu_0 H_{K,eff}$, extracted using planar and anomalous Hall effect measurements, decreases (Fig. 3b), indicating a decrease in in-plane anisotropy (see "Magnetic characterization" for details concerning the extraction of $\mu_0 H_{K,eff}$). At a fluence of $2.45\,J\,cm^{-2}$,

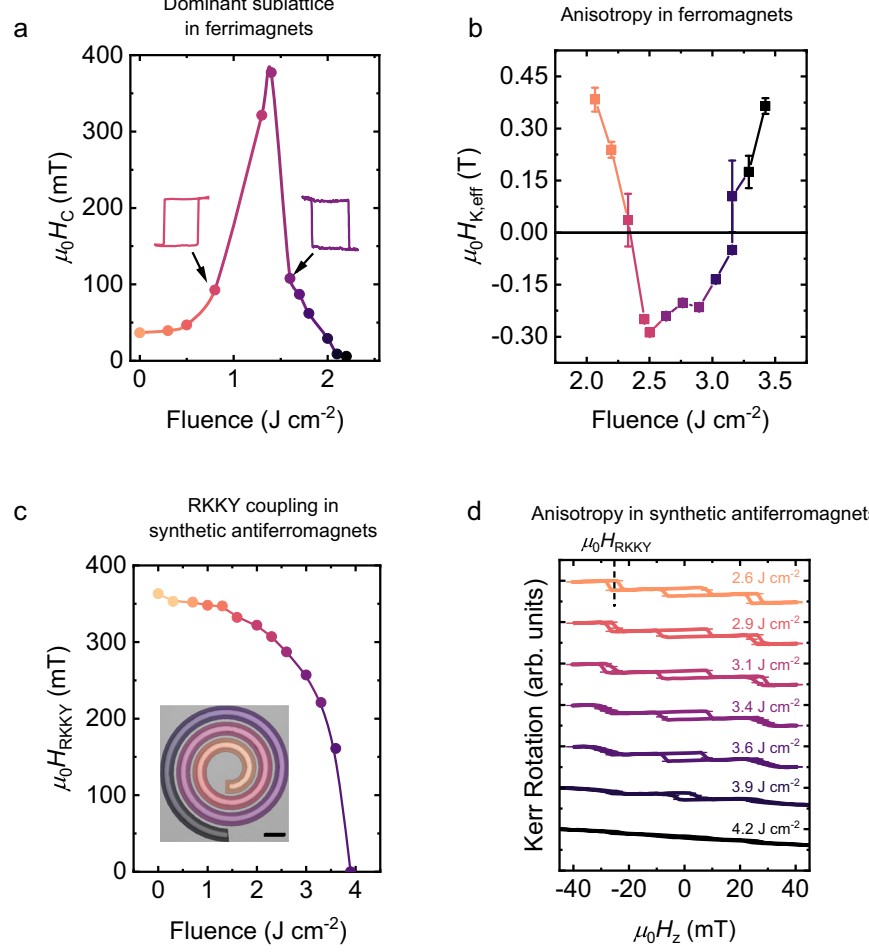

**Fig. 3 | Tuning of magnetic properties with DWLA. a** Coercive field $\mu_0 H_C$ as a function of DWLA laser fluence for a CoGd ferrimagnet, determined using pMOKE microscopy. The uncertainties in the $\mu_0 H_C$ values are determined by the field step size used during each measurement and are smaller than the symbol size (see "Magneto-optic Kerr effect measurements" for details). The insets show representative hysteresis loops collected using pMOKE microscopy for fluences on either side of the divergence in $\mu_0 H_C$. Hysteresis loops for all laser fluences are provided in Supplementary Fig. S9. **b** Effective magnetic anisotropy for CoFeB films as a function of laser fluence. PMA (negative $\mu_0 H_{K,eff}$) is observed between 2.45 J cm$^{-2}$ and 3.15 J cm$^{-2}$. Error bars represent the standard deviation of $\mu_0 H_{K,eff}$ from fitting the data (see "Magnetic characterization"). **c** Strength of antiferromagnetic RKKY exchange coupling field $\mu_0 H_{RKKY}$ as a function of DWLA laser fluence for a Co/Cr/Co SAF, determined using pMOKE microscopy. The uncertainties in the $\mu_0 H_{RKKY}$ values (determined by the field step size used during each

measurement) are smaller than the symbol size. The inset demonstrates the structure of the sample and the laser fluence profile, showing how dramatic gradients in properties can be written within small areas. The fluence color scale matches that used in the main figure. Scale bar = 10 μm. **d** CoFeB/Pt-based SAFs with a Ru spacer layer display a spin-flip transition as-grown and when exposed to low laser fluences (see orange curve, 2.6 J cm$^{-2}$), where the two magnetic layers abruptly switch from antiferromagnetic to ferromagnetic alignment with a square loop centered about $\mu_0 H_{RKKY} \approx 25$ mT, indicated by the dashed line. As the laser fluence increases, the SAF instead displays a more gradual spin-flop transition from antiferromagnetic to ferromagnetic alignment (see purple curve, 3.6 J cm$^{-2}$). Throughout Fig. 3, darker colors correspond to higher laser fluences as a guide to the eye, while the range of laser fluences is defined by the fluence axis or fluence values listed within each panel. The uncertainty in the Kerr rotation was determined using the approach stated in "Magneto-optic Kerr effect measurements".

$\mu_0 H_{K,eff}$ becomes negative, corresponding to a change of the magnetic easy axis from in-plane to out-of-plane. When the laser fluence is increased above 3.15 J cm$^{-2}$, $\mu_0 H_{K,eff}$ becomes positive again, returning to in-plane magnetic anisotropy. These results are consistent with those obtained by furnace annealing CoFeB films, where there is an optimal temperature range of ~200 °C – 350 °C that promotes strong PMA[45]. We confirm that laser annealing and furnace annealing give similar effects by comparing films subjected to each treatment (see magnetic hysteresis loops in Supplementary Fig. S10). However, laser annealing provides local control impossible to achieve with furnace annealing. Rather than a uniform transformation of properties, an anisotropy energy range of up to $\Delta K_{eff} \approx 3 \times 10^5$ J m$^{-3}$ can be created in arbitrary microscale patterns in the film plane.

Now, we show how the laser annealing-induced interfacial interdiffusion of Co/Cr/Co SAFs modifies the interlayer exchange, or Ruderman-Kittel-Kasuya-Yosida (RKKY), coupling strength. To

demonstrate how the unique two-dimensional patterning with DWLA allows for the creation of dramatic gradients in magnetic properties within small areas, we consider the impact of a spiral-shaped fluence gradient applied to a magnetic structure, as shown in the inset of Fig. 3c. The as-deposited Co/Cr/Co SAF sample exhibits a pMOKE hysteresis loop typical of SAFs with PMA (see Supplementary Fig. S11a). From this pMOKE hysteresis loop, we can define the RKKY coupling field $\mu_0 H_{RKKY}$, which reflects the RKKY coupling strength, as the magnetic field about which the fields associated with the two spin-flip transitions between antiparallel and parallel alignment are centered. The as-grown sample possesses a $\mu_0 H_{RKKY}$ of 360 mT. From pMOKE hysteresis loops recorded from areas along the spiral structure subjected to laser fluences in the range 0 J cm$^{-2}$ to 4.5 J cm$^{-2}$, we find that there is a decrease in $\mu_0 H_{RKKY}$ as a function of laser fluence, as shown in Fig. 3c (see Supplementary Fig. S11a for pMOKE loops at these fluences). At a fluence of approximately 4 J cm$^{-2}$, a spin-flip transition is no

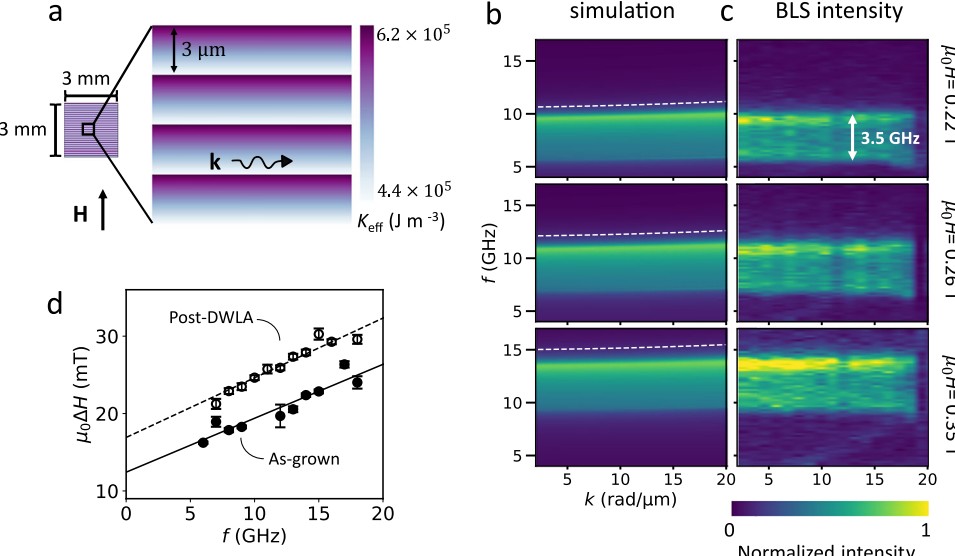

**Fig. 4 | Graded CoFeB magnonic channels. a** Top-down view of a CoFeB/MgO film containing multiple 3 μm-wide stripes with linear magnetic anisotropy gradients written over a large area of 3 mm × 3 mm. **b** Spin wave dispersion of the patterned film from mumax3 simulations. The white dashed line indicates the dispersion of the as-grown film. **c** Spin wave dispersion of the patterned film from k-resolved BLS measurements. A 3.5 GHz band of allowed frequencies is observed for the patterned film at all applied magnetic fields (indicated by the white arrow in the top panel). With an applied magnetic field of $\mu_0 H = 0.22$ T, 6 GHz – 9.5 GHz is

allowed (top panel). As the field increases to 0.26 T, the frequency band shifts to 7.3 GHz – 10.8 GHz (middle panel), and then to 9.8 GHz – 13.3 GHz at $\mu_0 H = 0.35$ T (lower panel). **d** Out-of-plane ferromagnetic resonance linewidth $\mu_0 \Delta H_{FMR}$ vs frequency $f$ for an as-grown and annealed film. The Gilbert damping (proportional to the slope of the linear relationship between linewidth and frequency) is comparable for the as-grown and annealed film. Error bars indicate the standard deviation of the linewidth from fitting the data (see "Magnetic characterization").

longer apparent in the pMOKE hysteresis loops (see Supplementary Fig. S11a), suggesting that the two FM layers comprising the Co/Cr/Co SAF structure are no longer antiferromagnetically coupled to each other, to which we assign a $\mu_0 H_{RKKY}$ value of zero. We believe this is due to the Cr spacer layer becoming magnetic by interdiffusion (see Supplementary Fig. S8), transforming the Co/Cr/Co SAF structure to that of a ferromagnetic heterostructure[55,56]. This ability to tune the RKKY coupling strength is achieved while leaving the saturation magnetization relatively unchanged (see Supplementary Fig. S11b). This indicates that each ferromagnetic layer has magnetic properties similar to those of the layers of the as-grown film, and the change in interlayer exchange coupling is due to a change in the spacer layer properties.

Finally, by implementing different materials in a SAF heterostructure, namely CoFeB/Pt/Ru, interdiffusion with DWLA can be leveraged to modify the magnetic anisotropy of the SAF while leaving the RKKY coupling unchanged. Two CoFeB/Pt magnetic multilayers separated by a Ru spacer are synthesized for this purpose. It has been found that PMA is lost in CoFeB/Pt multilayers when the structure is annealed above ~300 °C[43] due to interdiffusion that disrupts the interfacial anisotropy. Ru spacer layers are typically stable up to ~400 °C[57], so there is a process window where the CoFeB/Pt is affected, but the Ru is not. Interdiffusion indicated by XRD measurements given in Fig. 2c suggests that laser annealing should produce a similar decrease in PMA. Indeed, as the laser fluence is increased from 2.6 J cm$^{-2}$ to 3.9 J cm$^{-2}$, we observe a change in the shape of the pMOKE loop from that of a spin-flip SAF, where there is sharp switching centered at $\mu_0 H_{RKKY}$, to that of a spin-flop SAF, where switching from antiferromagnetic to ferromagnetic alignment occurs gradually, reflecting weaker PMA (see Fig. 3d). Despite the change in shape of the pMOKE loop, $\mu_0 H_{RKKY}$ does not change with laser fluence. We conclude that the interdiffusion within each magnetic layer leads to a change in magnetic anisotropy of the CoFeB/Pt/Ru SAF as the laser fluence increases – from strongly out-of-plane, to weakly out-of-plane, to in-plane, without disrupting the

RKKY coupling. This is confirmed with magnetometry measurements (see Supplementary Fig. S7).

### Large-area tunable anisotropy gradients for magnonics

To demonstrate the utility of 2D gradients in magnetic properties for magnonics applications, a large-area graded index design is fabricated in CoFeB/MgO films. For this, a 3 mm × 3 mm area is laser annealed with repeating 3 μm-wide linear gradient stripes, as shown in Fig. 4a. The laser fluence range is chosen such that $K_{eff}$ varies from $4.4 \times 10^5$ J m$^{-3}$ to $6.2 \times 10^5$ J m$^{-3}$ (corresponding to a change in anisotropy field of $\mu_0 H_{K,eff} = 0.21$ T to $\mu_0 H_{K,eff} = -0.15$ T) across the stripes. The spin wave dispersion, relating the spin wave wavelength (given by $2\pi/k$) to the frequency $f$ for a given applied magnetic field, is measured with wavevector-resolved Brillouin light scattering (BLS) spectroscopy and simulated with mumax3[58,59] (simulation details can be found in Section S4.1 and Supplementary Fig. S12 of the Supplementary Information). We characterize the Damon-Eshbach mode, where the spin waves propagate perpendicular to the applied magnetic field (schematically shown in Fig. 4a). The wavevector-resolved BLS measurement does not have spatial resolution, so the signal is collected from a >30 μm laser spot, covering multiple repetitions of gradient stripes. The resulting dispersion at three fields is shown for simulations in Fig. 4b and for BLS measurements in Fig. 4c.

The as-grown film has a single spin wave mode, shown by the dashed white lines for the three different applied magnetic fields in Fig. 4b. After creating the graded stripe pattern with DWLA, a broad plateau of intensity from 6 GHz to 9.5 GHz is measured with BLS at $\mu_0 H = 0.22$ T, seen in the top panel of Fig. 4c. This suggests that these graded anisotropy stripes act as a band pass filter for spin waves, allowing only spin waves in a 3.5 GHz range to propagate, and these frequencies can be tuned with the applied magnetic field to 7.3 GHz – 10.8 GHz at $\mu_0 H = 0.26$ T and 9.8 GHz – 13.3 GHz at $\mu_0 H = 0.35$ T, seen in the middle and lower panels of Fig. 4c, respectively. Furthermore, micromagnetic simulations (Fig. 4b) reproduce the BLS data with high fidelity (see Supplementary Fig. S13 for a comparison at specific

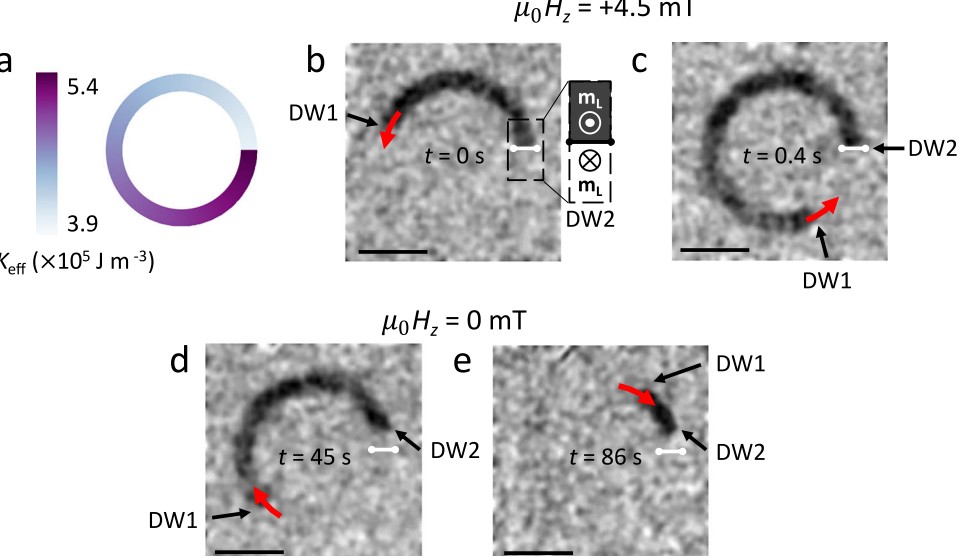

**Fig. 5 | Magnetic domain wall motion along a circular anisotropy gradient in a SAF. a** Circular magnetic anisotropy gradient written into a continuous CoFeB/Pt/Ru SAF, giving a radial domain wall ratchet. The track width is 3 µm, and the outer diameter of the track is 26 µm. **b**−**e** Background-subtracted Kerr microscope images of the device (scale bar = 10 µm). **b** The device is shown just at the moment a $\mu_0 H_z$ = +4.5 mT magnetic field is applied. A domain nucleates at the lowest anisotropy area (indicated with a white dumbbell), with the domain wall configuration shown to the right. Here, $m_L$ refers to the magnetization of the lower magnetic layer in the SAF, which determines the Kerr contrast. **c**, The device shortly after applying a $\mu_0 H_z$ = +4.5 mT magnetic field pulse. DW1 travels along the gradient, while DW2 remains pinned at the energy barrier between low and high magnetic anisotropy areas. **d** 45 s after the applied magnetic field is returned to zero, DW1 has started retreating back down the anisotropy gradient. DW2 moves slightly, possibly due to sample inhomogeneity, giving a small area where the equilibrium magnetization is in-plane. **e** 86 s after the field pulse, DW1 has continued to move spontaneously down the anisotropy gradient towards the white dumbbell where the domain originally nucleated. The background was taken at remanence after application of a $\mu_0 H_z$ = −30 mT magnetic field. The red arrows indicate the direction of domain wall motion. As the Kerr contrast of SAFs at remanence is small, the contrast was enhanced with post-processing explained in "Magneto-optic Kerr effect measurements". Unaltered images are shown in Supplementary Fig. S16.

wavevectors), reflecting the ability to precisely fabricate a desired magnetic anisotropy landscape in films. To verify that the designed gradient pattern in the anisotropy was accurately reproduced in the film, we have also compared simulation and BLS data for spin wave propagation orthogonal to the gradient stripes in Supplementary Fig. S14. If a narrower filter is desired (1.5 GHz bandwidth, for example), this can be constructed by writing shallower gradients (see Supplementary Fig. S15).

To date, gradient magnonic devices have been made with ion irradiation, which can damage films and increase magnetic damping at higher doses[16]. With laser annealing, the film quality is maintained, and can even be improved (for example, the emergent crystallinity seen by comparing the TEM images in Fig. 2a). For the implementation of graded magnonic crystals in devices, it is necessary for spin waves to interact with changes in magnetic properties without increased dissipation, given by the Gilbert damping parameter. By measuring the ferromagnetic resonance linewidth $\mu_0 \Delta H_{FMR}$ at a number of excitation frequencies (Fig. 4d), we obtain a magnetic Gilbert damping for the as-grown film of $\alpha$ = 0.018 ± 0.002, while for a uniformly annealed film, $\alpha$ = 0.020 ± 0.002 (see "Magnetic characterization"). This consistent damping despite a change of magnetic anisotropy ($\mu_0 \Delta H_{K,eff}$ = 0.1 T) makes DWLA promising for applications in magnonics, where maintaining low damping is essential for high-performance devices.

## 2D anisotropy gradients for spintronics

The manipulation of magnetic domain walls with electric currents is important for next-generation in-memory computing. For example, recent computational work has shown how the spontaneous motion of domain walls in response to a gradient in anisotropy energy can be implemented as an artificial neuron[60]. SAFs are particularly well-suited

for this purpose, as antiferromagnetic systems are predicted to have high spontaneous domain wall velocities[23], enabling observation of spontaneous domain wall motion on the µm scale in the time span of several seconds. Here, we construct a 2D gradient in magnetic anisotropy in a synthetic antiferromagnet, creating a device that has been theoretically proposed[23], but has not been realized due to a lack of appropriate nanofabrication methods. We experimentally demonstrate the predicted spontaneous domain wall motion and observe it over micrometer-scale distances.

We fabricate a 3 µm-wide circular domain wall track in a CoFeB/Pt/Ru SAF where the laser fluence linearly increases along the track (see Fig. 5a). It is not possible to fabricate such a structure with most other methods except ion irradiation, which suffers from difficulty in maintaining constant doses radially[22]. The linear increase in laser fluence leads to a linear decrease in the PMA of the CoFeB/Pt/Ru SAF, with $\Delta K_{eff}$ = 1.5 × 10⁵ J m⁻³ along the 72 µm average circumference of the circle (see hard-axis hysteresis loops in Supplementary Fig. S7b). The bottom magnetic layer of the SAF structure has a larger magnetization than the top layer so that the Kerr contrast observed corresponds to the magnetic state of the lower magnetic layer. The device is initialized with a -30 mT out-of-plane magnetic field to a down-up state, referring to the magnetization direction in the bottom and top magnetic layer, respectively. When a +4.5 mT magnetic field is applied, an up-down domain (dark contrast in Fig. 5b) nucleates in the lowest anisotropy area (indicated by the white dumbbell in Fig. 5b−e), and one domain wall (DW1) propagates counterclockwise along the circular anisotropy gradient (Fig. 5c). Due to the anisotropy energy barrier, the other domain wall (DW2) is prevented from moving clockwise, as seen in Fig. 5b, c. When the field is removed, DW1 retreats to the area with the lowest anisotropy, moving spontaneously down the anisotropy gradient at an average speed of ~580 nm s⁻¹ (based on the total distance traveled by the domain wall in the observed 86 s), as seen in Fig. 5d, e.

Such a resetting functionality can fulfill one of the crucial criteria for an artificial neuron in neuromorphic computing devices. Furthermore, these curved gradient tracks can be implemented as a feedback mechanism for domain wall logic, so the output of a logic gate can be fed back into an input. This is essential for the development of in-memory computing[61].

## Future directions

We have demonstrated the creation of 2D gradients of the chemical and physical microstructure in a wide variety of thin film systems. These gradients, with arbitrary profiles and directions, can be used to obtain complex, continuously varying patterns in the functional material properties, achieved by combining the tunability of laser annealing with the design flexibility available with 2.5D photolithography systems. Modification of all layers in a heterostructure is possible because the laser deposits thermal energy that penetrates tens of nanometers into materials, and the large fluence range enables desired changes over a wide range of materials, structures, and underlying physical changes. Utilizing photolithography equipment allows us to make complex designs with 180 nm feature sizes (see Supplementary Fig. S17) at high speeds of up to 3 mm²/minute. The high speed of DWLA also means that mm to cm-scale areas can be exposed with high uniformity. It is further anticipated that using multiple laser sources in parallel would facilitate even faster exposure times.

By designing intricate magnetic energy landscapes to create new material functionality, we have enabled two applications in magnetic domain wall computation and magnonics that could not be realized using existing methods. Such fine control of the energy landscape, both in terms of creating complex patterns and grayscale precision of laser fluence, will lead to new schemes for in-memory computation, data storage, and sensing. Beyond the field of magnetism[62,63], the creation of 2D gradients in the properties of any thin film that transforms in response to heat can lead to further applications across electronics, photonics, microfluidics, and any other application that would benefit from complex designs of gradients in material properties at the mesoscopic scale[64].

## Methods

### Synthesis

All films were deposited on Si substrates with a 300 nm-thick thermal oxide coating (referred to as SiOx) using magnetron sputtering in a 0.4 Pa Ar partial pressure. For this, we used an ATC Orion sputtering tool (AJA International, Inc.), which had a base pressure of $2 \times 10^{-6}$ Pa. MgO and Ta were deposited with RF sputtering, while all other layers were deposited with DC sputtering. The full compositions of the films are detailed below from the lowest layer next to the substrate to the uppermost layer in each stack, where all thicknesses are in nm:

- CoFeB/MgO ferromagnetic films, with the structure Ta(5)/$Co_{40}Fe_{40}B_{20}$(1.3)/MgO(1.5)/Ta(5) were sputtered at powers of 80 W for Ta, 40 W for CoFeB, and 150 W for MgO.
- CoGd ferrimagnetic films with the structure Ta(2)/Pt(5)/$Co_{70}Gd_{30}$(5.5)/Ta(3) were deposited using a co-sputtering technique, in which the Co, Gd, Ta, and Pt were deposited using sputtering powers of 50 W, 24 W, 100 W, and 100 W, respectively. The subscripts used to specify the stoichiometry of the CoGd layer correspond to atomic percentages.
- The Co/X/Co (X = Cr or Ta) SAFs have the composition Ta(2)/Pt(3)/Co(0.7)/X/[Co(0.7)/Pt(1)]$_3$/Pt(5), with X being Cr(1) or Ta(1.25). All layers were deposited using a sputtering power of 100 W.
- The CoFeB/Pt/Ru SAFs have the structure Ta(5)/[Pt(1.5)/$Co_{40}Fe_{40}B_{20}$(0.4)]$_3$/Ru(1.1)/[$Co_{40}Fe_{40}B_{20}$(0.4)/Pt(1.5)]$_2$. Ta and Ru were deposited at a power of 80 W, CoFeB was deposited at 40 W, and Pt was deposited at 100 W.

After treatment with DWLA, the Co/Cr/Co SAF film was patterned into a 2 µm-wide spiral feature (as shown in Fig. 3c) using standard UV photolithography and ion beam etching techniques.

### Direct-write laser annealing

A Heidelberg DWL66+ direct-write photolithography system with 405 nm wavelength continuous wave (CW) illumination was used to perform DWLA (schematic shown in Supplementary Fig. S17a). In the "Advanced Grayscale" configuration, this platform provides 256 levels of grayscale exposure intensity, which allows for quasi-continuous variations in laser fluences. Design files made using commercial CAD programs modulate the power of the laser as it is raster-scanned over the film. The "Hi-Res" write head used throughout this work offers a 50 nm address grid with a minimum feature size of 180 nm (see Supplementary Fig. S17b). The maximum laser fluence possible is ~31 J cm⁻². Based on the published write speed of 3 mm²/minute, we estimate that each pixel in the address grid is exposed to the laser for 50 ns for any exposure.

From previous reports on furnace annealing work, we know that the different heat-induced physical changes observed in our materials (crystallization, interdiffusion, and oxidation) occur at different temperatures. For example, the electronic structure of rare-earth elements gives Gd a strong affinity for oxidation[65], and thus the effects of oxidation are observed after furnace annealing films at temperatures as low as 100 °C[44]. As such, when designing a property gradient in functional thin films, one must consider the energy barrier for a particular physical or chemical transformation, which will in turn define the laser fluences needed to obtain the desired changes in the magnetic properties.

The thickness of a film and the substrate on which it is grown can dramatically impact the temperature change in response to a given laser fluence and the volume within the film that experiences a significant temperature change. From our simulations of heating due to laser annealing (see Supplementary Fig. S1), we can conclude the following:

(i)   For the same laser exposure parameters, the average temperature increase of a metallic film is inversely proportional to the thermal conductivity of the substrate used.

(ii)  The lateral region of elevated temperature extends over a larger area when the metallic film is on a substrate with high thermal conductivity.

(iii) Thinner metallic films exhibit a more uniform temperature through their thickness in response to localized surface heating.

Further details regarding the thermal simulations can be found in the "Modeling of thermal properties" subsection below and Supplementary Fig. S1.

A silicon substrate with a 300 nm-thick thermal oxide (SiOx) coating was used for all experiments, as the low thermal conductivity of SiOx (1.3 W m⁻¹ K⁻¹) reduces the laser fluence required to obtain significant temperature increases and gives a better localization of the heat to the laser spot position (see Supplementary Fig. S1). Additionally, we note that the maximum fluence of ~31 J cm⁻² offered by the DWL66+ makes DWLA suitable for patterning the physical and magnetic properties of films on substrates with higher thermal conductivity, lower optical absorptivity, and lower absorption of laser light than that of silicon substrates, and for targeting physical transformations with high activation energies.

### Magneto-optic Kerr effect measurements

Because the magnetic property changes in response to annealing manifest themselves in the out-of-plane magnetic response of our films, we collected hysteresis loops in the polar MOKE geometry to determine how the magnetic properties change with increasing laser fluence. The

polar magneto-optic Kerr effect (pMOKE) hysteresis loops were obtained using two different systems: A MOKE magnetometer (optimized for high signal-to-noise hysteresis loop measurements) and a MOKE microscope (tailored for imaging). The technique used to collect the data extracted from the hysteresis loop measurements is stated in the figure captions. For more complex magnetic patterns induced by DWLA, the collection of location-specific hysteresis loops was easier using the MOKE microscope, where identifying the region of interest is more straightforward, although the signal-to-noise ratio is poorer than that of the MOKE magnetometer. Similarly, depending on the physical transformation induced by DWLA, the associated change in the optical reflectivity of a material can be subtle, making it easier to discern using a MOKE microscope optimized for imaging.

The MOKE magnetometer employed was a NanoMOKE system manufactured by Durham Magneto Optics, Ltd with 660 nm wavelength laser illumination. A MOKE microscopy platform from Evico Magnetics GmbH featuring wide-spectrum LED illumination was used for imaging. Each system allows for the collection of location-specific hysteresis loops from areas that were annealed using the fluences indicated in the figures. For the ferrimagnets and CoFeB/Pt-based SAFs, a series of 50 μm × 50 μm boxes (with ~200 μm spacing between boxes) was annealed with different laser fluences. The spiral structure shown in Fig. 3c, with a laser fluence gradient along the spiral, was used to characterize magnetic changes of the Co/Cr/Co SAF.

When collecting images of domain states using MOKE microscopy, the contrast of the captured images was enhanced using a background subtraction procedure in which an image collected on application of a saturating magnetic field was subtracted from the recorded image, as is common in MOKE microscopy. The setting of the analyzing polarizer of the MOKE microscope was adjusted to maximize the extinction of reflected light with the same polarization as the incident light, and such that dark (light) contrast corresponds to up (down) magnetization. The pMOKE hysteresis loops for the ferrimagnets were obtained with the same analyzing polarizer setting. The wavelength of illumination used dictates that MOKE measurements of our CoGd ferrimagnets were primarily sensitive to the magnetization of the Co sublattice[66,67]. To obtain the MOKE microscope images in Fig. 5, a Gaussian blur was applied to the original images, and the contrast was enhanced. Unaltered images are shown in Supplementary Fig. S16.

The error of grayscale intensity values for hysteresis loops recorded with MOKE microscopy was determined by calculating the standard deviation of all normalized grayscale values above the sample's saturation field (at least 10 points) during each measurement. The uncertainties in the RKKY and coercive field values extracted from MOKE microscopy data are determined by the magnetic field step size used.

### Structural characterization

X-ray diffraction and reflectivity measurements were conducted using a Bruker D8 Discover, equipped with a microfocused Cu source. A Ni filter was used to partially monochromate the incident x-rays to the Cu Kα line (0.154 nm). The angular step size $\Delta 2\theta$ for diffraction and reflectivity measurements was 0.03° and 0.02°, respectively. Structural modeling of x-ray reflectivity curves was performed using GenX[68] (see Supplementary Fig. S2). The figure of merit for each model was < 0.1.

### Magnetic characterization

A Quantum Design magnetic property measurement system (MPMS) equipped with superconducting quantum interference device - vibrating sample magnetometry (SQUID-VSM) was used to measure the saturation magnetization of selected continuous films subjected to uniform laser exposure. The standard error of each

magnetization measurement, given by the standard error of all magnetic moment values (at least 26) recorded during the acquisition time, is $< 2 \times 10^{-4}$ A m$^2$, corresponding to <1% error for each data point. When extracting the saturation magnetization from SQUID-VSM, we report the standard deviation of the data points above the saturation field as the measurement uncertainty.

Effective anisotropy fields were determined from anomalous Hall effect (AHE) and planar Hall effect (PHE) measurements on a CoFeB film with Hall bar device patterns. For AHE measurements, a DC current is applied in the $x$ direction (in the film plane), a field is applied in the $z$ direction (out of the sample plane), and the voltage along the $y$ direction is recorded. $\mu_0 H_{K,\text{eff}}$ is then given by the field where the linear fit of the low-field anomalous Hall resistance and the linear fit of the high-field anomalous Hall resistance intercept.

For PHE measurements, a dc current and magnetic field are applied along the $x$ direction (in the plane of the sample), and the voltage along the $y$ direction is recorded as the field is swept from $\mu_0 H = 0.7$ T to $\mu_0 H = -0.7$ T. The low-field planar Hall voltage is fit to the equation[69]:

$$R_H = R_0 \sqrt{1 - \left(\frac{\mu_0 H_x}{\mu_0 H_{K,\text{eff}}}\right)^2} \tag{1}$$

where $R_0$ is a scaling factor and $\mu_0 H_{K,\text{eff}}$ is the effective anisotropy field, given by $\mu_0 H_{K,\text{eff}} = \mu_0(M_s + H_K^\perp)$ (where $H_K^\perp$ is the perpendicular anisotropy field). $\mu_0 H_{K,\text{eff}}$ is defined as positive when the film has in-plane anisotropy, and negative when the film has perpendicular anisotropy.

Out-of-plane ferromagnetic resonance measurements were performed on films to determine the Gilbert damping using a flip-chip setup with a coplanar waveguide. The applied DC magnetic field was swept with a fixed frequency RF field, and the change in microwave absorption with field was recorded. Data for each frequency is fit to a Lorentzian derivative peak, which provides the linewidth. The width of this peak is related to the Gilbert damping $\alpha$ of the film by the following equation:

$$\mu_0 \Delta H_{\text{FMR}} = \mu_0 \Delta H_0 + \frac{4\pi\alpha}{\gamma} f \tag{2}$$

where $\gamma$ is the gyromagnetic ratio and $\mu_0 \Delta H_0$ is the zero-frequency linewidth. The zero-frequency linewidth is directly related to the inhomogeneity in the magnetic properties of the film.

### Brillouin light scattering

We employed a standard Brillouin light scattering setup using a six-pass, tandem Fabry-Pérot interferometer. The incident laser with a wavelength of 532 nm and 60 mW power was focused onto the sample using a 50 mm lens. The lens was situated in an optical setup that enabled rotation of the sample in the Damon-Eschbach spin wave geometry with minimal movement of the laser spot on the sample surface and minimal change in focus from the lens. The sample was rotated between 5° and 77° ($k = 2$ μm$^{-1}$ − 3 μm$^{-1}$), and the setup provided an angular resolution of 0.002°.

### SIMS measurements

SIMS analysis was performed using an electrostatic quadrupole secondary ion mass spectrometer (Hiden Analytical EQS-MS). This was equipped with an Ar+ ion source at an ion energy of 1 kV (−35 nA ion beam current) and a sample area of 1 mm × 1 mm with 200 pixels × 200 pixels was scanned. The ion beam (with a diameter of 100 μm) followed a raster pattern during each etching cycle. Each etching cycle corresponds to the removal of a thin layer of the film, giving a depth profile. One quarter of the scanned area was selected as the analytical area. An electron flood source (FS40, PREVAC) was used

throughout the measurement for charge compensation. The CoGd ferrimagnetic films used for SIMS analysis had the same composition described in "Synthesis", but were grown on a Si substrate coated with 100 nm of LPCVD SiN to limit any $O^-$ ion contributions from the substrate.

## Modeling of thermal properties

The thermal response of metallic thin films was modeled using the "Heat transfer in solids" (ht) package of COMSOL Multiphysics. The temporal heat profile due to laser annealing was simulated by inputting thermal power to the top surface of thin films and observing the temperature changes induced through the film and substrate. The thermal properties of each material were defined using the materials library provided in COMSOL Multiphysics.

## Data availability

All data generated in this study have been deposited in the Zenodo database under accession code 10519203 https://doi.org/10.5281/zenodo.10519203.

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

## Acknowledgements

This work was supported by the European Union's Horizon 2020 research and innovation program under the Marie Skłodowska-Curie grant agreement No 884104 (PSI-FELLOW-III-3i) (JAB), the ETH Zurich Postdoctoral Fellowship Program 22-2 FEL-006 (LJR), and the Swiss National Science Foundation Ambizione scheme (216196) (NAS) and Projects scheme (204103) (XH). We thank Elisabeth Müller and the Electron Microscopy Facility at the Paul Scherrer Institut for transmission electron microscopy measurements. We thank J. Moritz Bosse for assistance with furnace annealing experiments. We thank the staff of the cleanroom facilities at the Laboratory for Nano and Quantum Technologies at the Paul Scherrer Institute for technical support. Certain trade names and company products are mentioned in the text or identified in an illustration in order to adequately specify the experimental procedure and equipment used. In no case does such identification imply recommendation or endorsement by the National Institute of Standards and Technology, nor does it imply that the products are necessarily the best available for the purpose.

## Author contributions

J.A.B. and L.J.R. synthesized samples, carried out experiments, performed the analysis, and conducted simulations. K.M. synthesized S.A.F. samples and carried out characterization experiments. X.H. and N.A.S. performed SIMS measurements. J.W. and H.T.N. carried out BLS measurements. J.A.B., L.J.R., A.H. and L.J.H. prepared the manuscript. All authors discussed the results and commented on the manuscript.

## Competing interests

The authors declare no competing interests.
