## [Transparent Peer Review file · Nature Communications]

Two-dimensional gradients in magnetic properties created with direct-write laser annealing

Corresponding Author: Dr Lauren Riddiford

Version 0:

Reviewer comments:

Reviewer #1

(Remarks to the Author)

I am disappointed that my recommendation to divide this work into separate papers was not adopted. The comprehensive nature of a review requires all questions to be addressed, and when multiple separate experiments are involved, this necessitates resolving each issue independently yet simultaneously—a considerably difficult task. I would advise focusing on a single material system. The current analysis of various material systems appears insufficient, lacking thorough quantitative discussion. To put it simply, the authors merely enumerate experimental results without providing meaningful insight into their significance or underlying mechanisms. The explanations seem to consist primarily of listing numerous possibilities. Unfortunately, as long as this structure combining three or four disparate experimental results persists, even if some experiments show promising results, the presence of less coherent experiments or discussions will undermine the overall cohesiveness of the research. I again recommend dividing this work into separate papers for submission to more specialized journals.

As an expert in laser annealing crystallization, magnetic materials, and spin waves, I find numerous perplexing aspects and insufficient evidence for drawing conclusions in these discussions, preventing me from recommending publication. The points raised below primarily indicate a lack of necessary data, which would require additional peer review after supplementary data is provided.

While some results not specifically addressed in my comments below may be valuable, I will refrain from commenting on those aspects.

Comments on Specific Sections

Section 1: Controlling Structural Transformations with DWLA

-. This section appears to be primarily introductory and instructional in nature. No claims of novelty are made, and indeed, I see no novel contribution here. The authors should clearly indicate which aspects are superior to or consistent with previous research. The connection between this section and subsequent discussions should also be explicitly established.

Section 2: Tuning Magnetic Properties with DWLA

-. Regarding Figure 3a, please clarify the validity of the magnetization measurements. The authors should provide a discussion of the coercivity magnitude, saturation magnetization values, and analyze the CoGd sublattice states. Without such context, it is difficult to assess the reliability of these results. The statement that "these behaviors are characteristic of ferrimagnets in the vicinity of a change in the dominant magnetic sublattice, which can be the result of a change in thickness, temperature, or in this case, chemical composition [44, 45]" seems insufficiently substantiated.

-. Concerning the CoFeB/MgO heterostructure results, previous literature has already demonstrated similar outcomes comparing laser annealing to furnace annealing in other material systems. Simply presenting the technique therefore lacks novelty. The authors should quantitatively demonstrate which specific aspects of their findings are definitively new. I do not currently discern significant novelty or interest in these results. The manuscript transitions to different material systems before thoroughly examining important aspects such as surface roughness changes, interface conditions, elemental diffusion at interfaces, and MgO insulating properties.

Section 3: Large-area Tunable Anisotropy Gradients for Magnonics

-. The computational model is not provided, making the calculation results incomprehensible.

- Figures 4a, 4b, and 4c lack clear explanation of what the different shades represent.
- What appears in Figure 4 seems to be spin wave propagation bands rather than band gaps. Band gaps typically appear within propagation bands. To properly discuss magnonic crystals introduced by laser-induced magnetization changes, structures with varying periodicity should be compared. Comparing differences when changing the laser gray scale gradient levels would also be valuable. Given the BLS technique used, mapping propagation characteristics in all directions would be useful for evaluating the effectiveness of this method. Overall, the claims, experiments, and analysis are not well aligned.
- Figures 4b and 4c cannot be considered equivalent. Spin wave propagation bands can be freely changed by the magnitude of the effective magnetic field, so this alone is insufficient to demonstrate agreement between experimental and calculated results. Several inconsistencies are puzzling: calculation spectra are curved while experimental spectra are straight; band gaps do not appear within propagation bands; and despite using BLS, evaluation is only done in specific directions. The configuration in Figures 4a, 4b, and 4c appears to be for surface waves, while Figure 4d seems to be for volume waves, which is also puzzling.

Reviewer #2

(Remarks to the Author)

I appreciate all the efforts made by the authors to address mine and other reviewer's comments, improve the readability, and, more importantly, add new data to make the manuscript more compelling and highlight novelty. I was also please to see the magnonic channel demonstration! I am therefore happy to recommend this paper for publication as is.

Reviewer #3

(Remarks to the Author)

The authors have addressed many of the concerns raised by all the reviewers in the first round of reviews, making the 'take-home message' of their manuscript clearer to the reader. Overall the manuscript is well written and the figures are well formulated. However, there are some issues that remain which should be addressed before it is ready to be published.

- 1) Some of the points that the authors stressed in their rebuttal letter as being addressed in the new manuscript are either only briefly touched upon, or somewhat hidden within the methods section of the manuscript. For example, the authors performed COMSOL simulations to determine materials parameters (affinity for oxidation, thermal conductivity of the substrate, layer thicknesses, etc...) which are important in determining the appropriate sample design to achieve a particular temperature profile, and therefore the desired magnetic properties. These design principles must be considered to fully realize the power of their technique, but are largely overlooked in the current main text of the manuscript, even though they have been explored by the authors.
- 2) The point of Figure 2 is to highlight the physical phenomena (crystallization, oxidation, and interdiffusion) which are the dominant mechanisms in each material system investigated. How do the authors rule out that each case does not result from a combination of these 3 phenomena? Perhaps this could be addressed by elaborating more on my concern in point 1. Some of the materials characterization provided is perhaps a bit lacking in rigor. The appearance of diffraction spots in the electron diffraction patterns is hard to see, it seems like a rather subtle effect, the authors only show a small range of 2θ values in the XRD scans – why was this range chosen – because this is the range where the most intense diffraction peaks for all the phases are observed? Or would we learn more by seeing a wider range of 2θ values presented in the extended material? Could SIMS analysis be performed for the interdiffusion samples to show the depth profiles are changed by the intermixing? That seems like the more straightforward method for demonstrating interdiffusion.
- 3) While the authors have removed one type of material system from their manuscript, there are still four systems to keep track of. I recommend that the authors use consistent notation throughout the manuscript to make sure it is clear to the reader which system is being discussed. In particular, different notations are used for the Co/Cr and Co/Ta-based heterostructures, which are also referred to as SAFs, and Co/Cr/Co films. The authors should also clarify if Si/SiO₂ substrates were used throughout the figures (as stated in the methods section) or whether the CoGd films were grown in SiN substrates as indicated in the figure caption to Figure 2.

Version 1:

Reviewer comments:

Reviewer #1

(Remarks to the Author)

Finally, content has been provided that reveals the results. I believe sufficient information has been provided for each experiment. I would have preferred this information to be provided from the beginning. I think it has reached a level suitable for publication.

Reviewer #3

(Remarks to the Author)

The authors have addressed the concerns from the reviewers about the lack of data to support their claims about what the laser technique is doing to the samples. I still have some concerns about the large number of samples which are described

in the manuscript, with the majority of the real 'science' relegated to the supplemental material.

We are grateful to the reviewers for taking the time to read our updated manuscript and for offering constructive feedback. For the reviewers' convenience, we have included a version of the revised main text with our changes tracked in our resubmission.

Based on feedback from the reviewers, we have added new supplementary data to quantitatively support our conclusions associated with the physical transformation mechanisms and to substantiate our comparison of micromagnetic simulations and BLS data for our magnonics application. Furthermore, we have significantly reorganized the Supplementary Information to provide structure and clarity as to which materials and physical properties are being discussed. The new section titles and, where relevant, the major new content additions are as follows:

- **Section S1: Modeling laser-induced heating of thin films**

- **Section S2: Clarifying competing physical transformations in material systems**
 - **Section S2.1: CoFeB Ferromagnets**
 - XRR data for CoFeB/MgO ferromagnetic films after DWLA exposure using various laser fluences, indicating that the interfacial roughness is not affected by the annealing process.
 - SQUID-VSM characterization demonstrating that the saturation magnetization is not impacted by DWLA, indicating that the films are not significantly oxidized during DWLA.
 - **Section S2.2: CoGd Ferrimagnets**
 - XRD measurements indicating that the crystalline structure of our CoGd films is unaffected by DWLA.
 - **Section S2.3: CoFeB/Pt/Ru SAFs**
 - Wide-range XRD measurements demonstrating that no additional structural phases are induced by DWLA.
 - **Section S2.4: Co/X/Co (X = Cr, Ta) SAFs**
 - SIMS characterization of a Co/Cr/Co-type structure before and after DWLA, indicating intermixing of the Co and Cr layers
 - XRD characterization of a Co/Ta/Co-type structure exhibiting no appreciable change in the crystalline structure after DWLA.

- **Section S3: Further magnetic characterization of films**

- **Section S4: Spin-wave propagation in CoFeB**
 - Details of the micromagnetic simulations of spin wave propagation, including comparisons between experimental versus simulated data for specific wavevectors and data on propagation of spin waves orthogonal to the gradient stripes.

- **Section S5: Unprocessed images of domain wall motion in CoFeB/Pt/Ru SAFs**
- **Section S6: Further characterization of DWLA**

We respond point by point to the reviewers' specific comments below. We believe that, with these changes, our manuscript is ready for publication in Nature Communications.

RESPONSE TO REVIEWERS' COMMENTS

Reviewer #1 (Remarks to the Author):

I am disappointed that my recommendation to divide this work into separate papers was not adopted. The comprehensive nature of a review requires all questions to be addressed, and when multiple separate experiments are involved, this necessitates resolving each issue independently yet simultaneously—a considerably difficult task. I would advise focusing on a single material system. The current analysis of various material systems appears insufficient, lacking thorough quantitative discussion. To put it simply, the authors merely enumerate experimental results without providing meaningful insight into their significance or underlying mechanisms. The explanations seem to consist primarily of listing numerous possibilities. Unfortunately, as long as this structure combining three or four disparate experimental results persists, even if some experiments show promising results, the presence of less coherent experiments or discussions will undermine the overall cohesiveness of the research. I again recommend dividing this work into separate papers for submission to more specialized journals.

We carefully considered the suggestion to divide the work into separate papers, and it would certainly have been a valid approach. Our decision to keep this paper intact was based on conversations with researchers in the community who expressed interest in the breadth and generality of the technique, which can be readily applied to various materials targeting different heat-assisted material changes. To clarify this motivation, we have included a discussion of our design principles and the rationale behind our choice of these materials at the beginning of our Results section and in the discussion of structural transformations.

Throughout the paper, we have clarified the mechanism responsible for the physical transformation of our thin films, and we have added supplementary data to support a conclusive dominant mechanism of structural transformation. Finally, we have emphasized the significance of the stated results where appropriate. We hope that the Reviewer will understand our motivation to keep the manuscript in this form.

As an expert in laser annealing crystallization, magnetic materials, and spin waves, I find numerous perplexing aspects and insufficient evidence for drawing conclusions in these discussions, preventing me

from recommending publication. The points raised below primarily indicate a lack of necessary data, which would require additional peer review after supplementary data is provided. While some results not specifically addressed in my comments below may be valuable, I will refrain from commenting on those aspects.

Comments on Specific Sections

Section 1: Controlling Structural Transformations with DWLA

-. This section appears to be primarily introductory and instructional in nature. No claims of novelty are made, and indeed, I see no novel contribution here. The authors should clearly indicate which aspects are superior to or consistent with previous research. The connection between this section and subsequent discussions should also be explicitly established.

We agree with the reviewer that the “Structural Transformation” section of our manuscript primarily illustrates that DWLA can induce physical changes that can indeed occur in response to other treatments, including furnace annealing, ion irradiation, and localized oxidation. This section of our revised manuscript, whose content was guided by questions raised by all reviewers during the first round of review, now illustrates a key thesis of our work: DWLA offers a universal and widely available technique capable of enacting disparate physical transformations in different magnetic thin films and providing gradients in the magnetic properties in complex patterns. To clarify this point, we have included additional text to this effect in the introductory paragraph of this section. Furthermore, the “Structural Transformation” section has been modified to include additional citations to studies in which techniques other than DWLA were used to achieve similar physical transformations in magnetic thin films.

Section 2: Tuning Magnetic Properties with DWLA

-. Regarding Figure 3a, please clarify the validity of the magnetization measurements. The authors should provide a discussion of the coercivity magnitude, saturation magnetization values, and analyze the CoGd sublattice states. Without such context, it is difficult to assess the reliability of these results.

We agree with the reviewer that there are other possible routes by which the coercivity or net saturation magnetization of a TM-RE ferrimagnet could be modified without a variation in the saturation magnetization of one or both elemental sublattices. For example, if laser annealing increases the roughness of the ferrimagnet, this could lead to an increase in coercivity. However, under the reasonable assumption that the degree of roughness would scale with the laser fluence used, this scenario is incompatible with the nonmonotonic trend we observe in the coercivity as a function of laser fluence. Regarding the magnitude of the magnetic parameters we measure, we note that while the saturation magnetization does indeed approach zero in the vicinity of the magnetic compensation temperature, it is the trend in the coercive field, rather than precise magnitudes, that defines a divergence in the coercivity.

Furthermore, while a divergence in the coercivity or a minimum in the saturation magnetization are both highly emblematic of magnetic compensation in a ferrimagnet, we agree that such behaviors do not directly indicate a change in the magnetically dominant elemental sublattice. We can draw such a conclusion by focusing on the sign of the Kerr rotation observed during our MOKE hysteresis loop measurements. As detailed in the Methods section of our revised manuscript, MOKE measurements conducted using visible light are primarily sensitive to the magnetic orientation of the TM sublattice. As such, the fact that we observe a change in the MOKE loop polarity following application of laser fluences on either side of the divergence in coercivity, or a minimum in saturation magnetization, concretely illustrates that these trends are linked to a change in the magnetically dominant elemental sublattice. While our SIMS analysis does not differentiate between O^- ions originating from Co or Gd, RBS characterization of CoGd ferrimagnets has conclusively demonstrated that Gd oxidizes much more readily than Co [Z. Liu et al., PRB 107, L100412 (2023)] supporting our observation that DWLA-induced oxidation provides a means of tuning the Gd sublattice magnetization, and thus, the ferrimagnetic properties of our CoGd films.

Considering the above points, we have modified Figure 3a to include inset MOKE loops measured for laser fluences on either side of the divergence in coercivity. This addition demonstrates that the divergence in coercivity we observe with increasing laser fluence occurs concomitant to a change in the magnetically dominant sublattice. Furthermore, we have modified our discussion of the CoGd system in the “Tuning magnetic properties” section to denote which observed behaviors suggest a change in the dominant sublattice and which behaviors directly indicate such a change.

The statement that "these behaviors are characteristic of ferrimagnets in the vicinity of a change in the dominant magnetic sublattice, which can be the result of a change in thickness, temperature, or in this case, chemical composition [44, 45]" seems insufficiently substantiated.

To address the reviewer’s concerns, we have revised the referenced statement to include additional citations to studies where a change in the dominant magnetic sublattice of a ferrimagnet was inferred from features such as divergent coercivity, nonmonotonic trends in saturation magnetization versus temperature, or a reversal in MOKE loop polarity. We also cite examples where these behaviors arise from post-growth treatments, including ion irradiation, oxygen plasma, and furnace annealing. In line with our response to the reviewer’s previous comment, these additions clarify that the magnetic property variations we observe in response to DWLA are consistent with the known effects of post-growth processing (including oxidation) and can yield results comparable to those achieved by modifying the growth parameters, such as thickness or TM–RE proportionality.

Concerning the CoFeB/MgO heterostructure results, previous literature has already demonstrated similar outcomes comparing laser annealing to furnace annealing in other material systems. Simply presenting the technique therefore, lacks novelty. The authors should quantitatively demonstrate which specific

aspects of their findings are definitively new. I do not currently discern significant novelty or interest in these results.

We agree with the reviewer that the effect of laser annealing on CoFeB/MgO heterostructures has been demonstrated (with the effect on exchange bias reported in A. Sharma et al, IEEE Transactions on Magnetics (Volume: 55, Issue: 1, January 2019)). This heterostructure is one of the systems shown due to its familiarity, where the effects of thermal annealing on magnetic properties as a function of temperature are well-known. This justifies our understanding of the effect of laser annealing in our systems as a thermally driven transformation. Here, the transformation is not new, but the ability to generate gradients in the properties in complex designs over large areas and the demonstration of applications in magnonics is new. To emphasize the novelty in this particular system, we have rephrased the discussion in the “Physical transformation” section to clarify what is novel and what is already known.

The manuscript transitions to different material systems before thoroughly examining important aspects such as surface roughness changes, interface conditions, elemental diffusion at interfaces, and MgO insulating properties.

In the main text, we have highlighted the dominant transformation mechanisms in our system. We have now added further physical characterization and explanation to the Supplementary Information for all films, considering changes in the surface roughness and interdiffusion. X-ray reflectivity measurements of CoFeB films indicate minimal changes in interface quality over the range of fluences studied. Oxidation of the CoFeB would decrease the saturation magnetization. In the fluence range providing PMA, we do not observe any decrease in the saturation magnetization. At fluences above the range of PMA, we do see a decreased magnetization, which may indicate oxidation. In CoFeB/Ru/Pt- and Co/Cr/Co-based SAFs, we do observe interface interdiffusion, as shown in Sections S2.3 and S2.4 of the Supplementary Information, respectively. In CoGd ferrimagnets, as mentioned above, we do not observe a systematic decrease in the saturation magnetization as a function of laser fluence, which would suggest diffusion of Pt and Ta into the magnetic layer. The additional data that we have added to the Section S2 of the Supplementary Information provides a more thorough examination of these film properties.

We verified that our MgO is insulating by measuring the 2-point resistance of a continuous 2.5 nm MgO film. This two-point resistance was >10 MOhm over 1 mm distance, confirming its insulating state. Additionally, while the insulating nature of MgO is essential for magnetic tunnel junctions, we consider the electrical properties of our films to be outside the scope of this work.

Section 3: Large-area Tunable Anisotropy Gradients for Magnonics

- The computational model is not provided, making the calculation results incomprehensible.

The mumax3 code for the model will be provided in the Zenodo repository, and we have attached the code to the resubmission for the convenience of the reviewer. Additionally, a description of the computational model is now included in Section S4.1 of the Supplementary Information.

- Figures 4a, 4b, and 4c lack clear explanation of what the different shades represent.

Many thanks to the reviewer for pointing this out. Legends have now been added to Figure 4.

- What appears in Figure 4 seems to be spin wave propagation bands rather than band gaps. Band gaps typically appear within propagation bands. To properly discuss magnonic crystals introduced by laser-induced magnetization changes, structures with varying periodicity should be compared. Comparing differences when changing the laser gray scale gradient levels would also be valuable. Given the BLS technique used, mapping propagation characteristics in all directions would be useful for evaluating the effectiveness of this method. Overall, the claims, experiments, and analysis are not well aligned.

Indeed, this is a demonstration of spin wave propagation bands, rather than demonstrating a band gap. We would not expect a band gap for this geometry, given that the spin waves travel along the gradient stripes and thus do not experience a periodic change in magnetic properties as would be the case for a magnonic crystal. We agree that the tunability of this is important; we have simulated changing the gradient levels (see Supplementary Figure S15), but we did not create multiple samples in an effort to move forward with resubmission.

We are slightly confused by the reviewer's comment to evaluate spin wave propagation bands in more directions. We can interpret this either as that they would like to see spatially resolved measurements using micro-focused BLS, or that they would like to see the spin wave propagation orthogonal to the stripes. Concerning the first interpretation, we only performed wave vector-resolved BLS, which does not have the spatial resolution to measure spin waves propagating in different directions. Additionally, micro-focused BLS does not have the wave vector resolution required to see these propagation bands, as the wave vector resolution is defined by the collection angle of the objective lens, which is intrinsically very large considering its numerical aperture. We now mention in the main text that we have wavevector resolution but not spatial resolution.

Regarding the second interpretation of this comment, we have added data and micromagnetic simulation for the propagation of spin waves orthogonal to the gradient stripes into the Supplementary Information (see Section S4.2). Simulating this geometry is difficult because the environment is locally asymmetric but has translational symmetry. In the simulation, spin waves are excited from a 10 nm linear source and propagate from that region (shown schematically in the new Supplementary Fig. S12). However, in the BLS experiment, all the spin waves excited within the >30 um laser spot are measured. Thus, comparison between experiment and theory is less reliable compared to the geometry shown in the main text. We note there may be more appropriate modeling tools [O. Wojewoda et al, Phys. Rev. B 110, 224428 (2024)], which are beyond our expertise.

Nevertheless, we can still compare the simulated and measured spectra for this geometry. Qualitatively, we observe that there is a single defined peak for both simulation and experiment in contrast to the wide plateau seen for the other geometry. Quantitatively, the spin wave frequency is ~1.5 GHz higher than the measured spin wave frequency (shown in the left panel in the figure below). This offset can be corrected

for by adding a uniaxial in-plane anisotropy in addition to the uniaxial out-of-plane anisotropy. A uniaxial in-plane hard axis along the stripes with an anisotropy energy of $K_{ip}=4.3 \times 10^4 \text{ J m}^{-3}$ (implemented using a custom field in mumax3) shifts the simulated frequency peak to match that of the experimental data (shown in the right panel in the figure below). For a direct comparison, all spectra are at $k = 9.7 \text{ rad } \mu\text{m}^{-1}$ for $\mu_0 H = 0.35 \text{ T}$.

To justify the inclusion of a uniaxial in-plane anisotropy, we performed longitudinal MOKE of the gradient pattern created in a similar CoFeB sample, which reveals strikingly different hysteresis loops when applying the field along the stripes and orthogonal to the stripes (see figure below). This difference confirms that there is an in-plane anisotropy in addition to the out-of-plane anisotropy. This makes sense because the written pattern is also very different along the two in-plane directions. The asymmetry of the loops for positive and negative magnetic field is due to the limited magnetic field range in our MOKE magnetometer, which means we cannot fully saturate the film in-plane. Because of this, we cannot extract the value of the anisotropy field from these MOKE loops. Still, the hysteresis loops provide evidence that an in-plane anisotropy component is appropriate to include in simulations. Including this anisotropy term does not affect the spin wave dispersion along the stripes because there is no magnetization component along the hard axis in this case.

In conclusion, we believe that the frequency peak of the spin waves propagating orthogonal to the stripes is sufficiently captured by simulations. The same anisotropy landscape is used for simulations of spin waves traveling along and orthogonal to the stripes, and both simulations are comparable to experimental data. Thus, this provides additional evidence that this technique is effective in producing designed magnetic landscapes. The data and discussion above have been added to the Supplementary Information section 4.2 and Supplementary Fig. S14.

- Figures 4b and 4c cannot be considered equivalent. Spin wave propagation bands can be freely changed by the magnitude of the effective magnetic field, so this alone is insufficient to demonstrate agreement between experimental and calculated results. Several inconsistencies are puzzling: calculation spectra are curved while experimental spectra are straight; band gaps do not appear within propagation bands; and despite using BLS, evaluation is only done in specific directions.

Thank you to the reviewer for pointing out possible inconsistencies in the presented data. First, regarding the shape of the calculated vs measured spectra, this can be explained by our estimation of the exchange stiffness and the reduced signal-to-noise ratio at higher k values in BLS. We have updated our exchange stiffness to 10 pJ m^{-1} in simulations (a value experimentally measured in CoFeB films by J. Cho et al, JMMM 339, P. 36-39 (2013)), which now more accurately reflects the experimental dispersion. In addition, we compare the FFT/BLS intensity vs frequency for several fixed k values for both techniques, seen below. Here, it is observed that the simulation and experiment agree very well for all k values, with only small differences observed on the shoulders of the peaks. The reduced number of counts at high k values means the signal-to-noise ratio is lower, leading to less sharp transitions from the signal to background. In the experiment, we are also measuring a small spread of wavevectors due to the collection angle of the focusing lens, which might also slightly alter the shoulders of the measured peaks. Finally, the saturation magnetization of the measured sample could be marginally lower than that of the simulated sample. This would modify the dipolar contribution to the dispersion and flatten the bands. We have added the following data to Section S4.2 of the Supplementary Information (Supplementary Fig. S13), as we agree this quantitative comparison is important to the results.

As mentioned above, we have also added data and micromagnetic simulation for the propagation of spin waves orthogonal to the gradient stripes to the Supplementary Information.

The configuration in Figures 4a, 4b, and 4c appears to be for surface waves, while Figure 4d seems to be for volume waves, which is also puzzling.

In Figure 4d, the ferromagnetic resonance mode is measured, and the linewidth from this uniform mode is plotted as a function of frequency. Because in-plane ferromagnetic resonance measurements are susceptible to contributions from magnon scattering, leading to possible error in the extracted Gilbert damping value, the magnetic field was applied out-of-plane. This damping value is a material parameter that is also relevant to the propagation of surface waves.

Reviewer #2 (Remarks to the Author):

I appreciate all the efforts made by the authors to address mine and other reviewer's comments, improve the readability, and, more importantly, add new data to make the manuscript more compelling and highlight novelty. I was also please to see the magnonic channel demonstration! I am therefore happy to recommend this paper for publication as is.

Thank you to the reviewer for spending the time to review the revised manuscript. We appreciate their recommendation to publish the paper in Nature Communications.

Reviewer #3 (Remarks to the Author):

The authors have addressed many of the concerns raised by all the reviewers in the first round of reviews, making the 'take-home message' of their manuscript clearer to the reader. Overall the manuscript is well written and the figures are well formulated. However, there are some issues that remain which should be addressed before it is ready to be published.

1) Some of the points that the authors stressed in their rebuttal letter as being addressed in the new manuscript are either only briefly touched upon, or somewhat hidden within the methods section of the manuscript. For example, the authors performed COMSOL simulations to determine materials parameters (affinity for oxidation, thermal conductivity of the substrate, layer thicknesses, etc...) which are important in determining the appropriate sample design to achieve a particular temperature profile, and therefore the desired magnetic properties. These design principles must be considered to fully realize the power of their technique, but are largely overlooked in the current main text of the manuscript, even though they have been explored by the authors.

We thank the reviewer for this suggestion to enhance our take-home message to the reader. We have now moved the discussion of design principles to the main text, both at the beginning of the results section, and we have highlighted strategic materials choices in the "Physical transformation" section.

2) The point of Figure 2 is to highlight the physical phenomena (crystallization, oxidation, and interdiffusion) which are the dominant mechanisms in each material system investigated. How do the authors rule out that each case does not result from a combination of these 3 phenomena? Perhaps this could be addressed by elaborating more on my concern in point 1.

Indeed, we employed specific sample design principles to give a single dominant physical transformation in response to heat. For example, the diffusion of Ru as a function of annealing temperature has been the subject of many studies due to its use in MTJ structures. It is typically thermally stable up to 400 °C. In addition, CoFeB/Pt multilayers were found to have significantly decreased PMA after annealing at 350 °C. Thus, there is a process window where we can create interdiffusion of CoFeB/Pt without causing Ru interdiffusion. We have now added more detailed descriptions of how we made such choices to the physical transformations. We have also added new x-ray reflectivity and SQUID-VSM data and associated discussion to the Supplementary Information to complement our claim of a single dominant physical transformation in each material system.

In the case of the CoGd ferrimagnets, we acknowledge that a change in the degree of crystallinity of the layers of our heterostructure may accompany the oxidation detected in response to DWLA, which could also modify the magnetic properties. To probe any changes in the crystallinity of the structure, we performed XRD measurements of a CoGd film before and after treatment with DWLA. As is typical for amorphous CoGd ferrimagnets, the as-grown film only exhibits a broad peak corresponding to the (111) polycrystalline texture of the Pt seed layer and an intense, narrow peak corresponding to the Si (004) reflection of the substrate. After uniform DWLA treatment, no new peaks become apparent, indicating that the CoGd layer remains macroscopically amorphous. Furthermore, the width of the rocking curve collected about the Pt (111) reflection does not show a substantial change after DWLA treatment, signifying that the polycrystalline texture of these layers is not affected by the annealing process. Potential changes in the crystallinity of the RKKY spacer layer in our Co/X/Co (X = Cr, Ta) SAFs were also assessed by performing XRD on a Ta film before and after DWLA. As with the CoGd ferrimagnet, we observe no evidence of new crystalline phases or any change in the polycrystalline texture of Ta following DWLA.

The measurements discussed above have been incorporated as new Supplementary Figs. S2, S3, S5, and S8, and a discussion of the results with appropriate citations has been added to the oxidation portion of the “Physical Transformation” section of the main text.

Some of the materials characterization provided is perhaps a bit lacking in rigor. The appearance of diffraction spots in the electron diffraction patterns is hard to see, it seems like a rather subtle effect,

The reviewer is correct that we should add context to the TEM of CoFeB films. The electron diffraction spots are indeed weak, reflecting the textured nature of the CoFeB/MgO interface, which is typical in this heterostructure [J. Liu et al, Appl. Phys. Lett. 107, 232408 (2015)]. Nonetheless, this onset of crystallization is sufficient to impact the magnetic anisotropy dramatically. We have added this reference and a short description to the discussion of the TEM to explain our results better.

the authors only show a small range of 2theta values in the XRD scans – why was this range chosen – because this is the range where the most intense diffraction peaks for all the phases are observed? Or would we learn more by seeing a wider range of 2theta values presented in the extended material?

Yes, the XRD data is shown in the 2theta range where we observe diffraction peaks from the film. An XRD scan of a larger range is now added to the Supplementary Information as Supplementary Fig. S6, where the Si substrate peak is also visible at 2theta = 69.25 degrees. We have also added an explanation of why this range was chosen to the description of the XRD data in the main text.

Could SIMS analysis be performed for the interdiffusion samples to show the depth profiles are changed by the intermixing? That seems like the more straightforward method for demonstrating interdiffusion.

Based on the referee's suggestion, we performed SIMS characterization of a Co/Cr bilayer sample before and after DWLA treatment. The measurements show that, following DWLA, Co appears much earlier during the etching process, and there is a broad range of etch cycles over which strong signals from both Co and Cr are detected, indicating that DWLA causes Co to diffuse into the Cr layer. These measurements have been included in the revised Supplementary Information as Supplementary Fig. S8a.

3) While the authors have removed one type of material system from their manuscript, there are still four systems to keep track of. I recommend that the authors use consistent notation throughout the manuscript to make sure it is clear to the reader which system is being discussed. In particular, different notations are used for the Co/Cr and Co/Ta-based heterostructures, which are also referred to as SAFs, and Co/Cr/Co films. The authors should also clarify if Si/SiO₂ substrates were used throughout the figures (as stated in the methods section) or whether the CoGd films were grown in SiN substrates as indicated in the figure caption to Figure 2.

We have updated our notation for different films to be consistent throughout the text. Indeed, the only film grown on SiN was the CoGd film used for SIMS, because we wanted to avoid any oxygen signal from an SiO_x-coated substrate. However, all other films are grown on Si/SiO_x. This point is now clarified in the Methods section.

REVIEWERS' COMMENTS

Reviewer #1 (Remarks to the Author):

Finally, content has been provided that reveals the results. I believe sufficient information has been provided for each experiment. I would have preferred this information to be provided from the beginning. I think it has reached a level suitable for publication.

We appreciate the reviewer's suggestions to help make our results more convincing and thank them for their recommendation that the manuscript is ready to be published.

Reviewer #3 (Remarks to the Author):

The authors have addressed the concerns from the reviewers about the lack of data to support their claims about what the laser technique is doing to the samples. I still have some concerns about the large number of samples which are described in the manuscript, with the majority of the real 'science' relegated to the supplemental material.

We are happy to hear that the reviewer believes we have addressed the stated concerns. We chose to address many samples to demonstrate the versatility of the technique and, while much of the materials science can be found in the Supplementary Information, we have ensured that all results are referred to in the main text.